

# Fish species identification on low resolution—a study with enhanced super-resolution generative adversarial network (ESRGAN), YOLO and VGG-16

Subhrangshu Adhikary[1], Saikat Banerjee[2], Rajani Singh[3] and Ashutosh Dhar Dwivedi[4]

[1] Research and Development, Spiraldevs Automation Industries Pvt. Ltd., Raiganj, West Bengal, India
[2] Remote Sensing, Aerosys Defence and Aerospace Pvt. Ltd., Pune, Maharashtra, India
[3] Department of Digitalization, Copenhagen Business School, Copenhagen, Denmark
[4] Cyber Security Group, Department of Electronic Systems, Aalborg University, Copenhagen, Denmark

Corresponding author
Subhrangshu Adhikary,
subhrangshu.adhikary@spiraldevs.com

## ABSTRACT

An intelligent detection and recognition model for the fish species from camera footage is urgently required as fishery contributes to a large portion of the world economy, and these kinds of advanced models can aid fishermen on a large scale. Such models incorporating a pick-and-place machine can be beneficial to sorting different fish species in bulk without human intervention, significantly reducing costs for large-scale fishing industries. Existing methods for detecting and recognizing fish species have many limitations, such as limited scalability, detection accuracy, failure to detect multiple species, degraded performance at a lower resolution, or pinpointing the exact location of the fish. Modifying the head of a compelling deep learning model, namely VGG-16, with pre-trained weights, can be used to detect both the species of the fish and find the exact location of the fish in an image by implementing a modified You Only Look Once (YOLO) to incorporate the bounding box regression head. We have proposed using the Enhanced Super Resolution Generative Adversarial Network (ESRGAN) algorithm and the proposed neural network to amplify the image resolution by a factor of 4. With this method, an overall detection accuracy of 96.5% has been obtained. The experiment has been conducted based on a total of 9,460 images spread across nine species. After further improving the model, a pick-and-place machine could be integrated to quickly sort the fish according to their species in different large-scale fish industries.

## INTRODUCTION

Around 34,000 fish species exist worldwide, and about 250 fishes are being discovered yearly (*Manjarrés-Hernández et al., 2021*; *Ward & McCann, 2017*). The per capita fish consumption per year rose from 9.9 kg in the 1960s to 20.5 kg in 2018. According to a report, 67 million tonnes of fish were exported for 164 billion US dollars in 2018, significantly impacting GDP worldwide (*SOFIA, 2022*). In the same year, 179 million

tonnes of fish were landed, of which 88% were used for direct human consumption. The report further states that aquaculture had record-high fish production in 2018, producing 114.5 million tonnes of fish, with a total valuation of 263.6 billion USD. All of these statistics signify the importance of fish in multiple aspects. Although the fish industry is so important, it still lacks many advanced technologies that may significantly improve its efficiency (*Naylor et al., 2021*; *Knausgård et al., 2022*). One such technology can be an advanced fish species identification system to detect the species accurately. This technology can be used to build an automated species-wise fish segregation system to reduce manual labor in separating fishes according to their species (*Liu et al., 2022*). In a practical scenario, the model might need to detect fish species from many fishes captured within a single camera frame (*Lalasa, Srija & Kumar, 2024*). Given the limited resolution of a digital camera, some fishes might be constrained to a lower number of pixels, making it harder to detect by a deep learning classification model (*Risholm et al., 2022*; *Jareño et al., 2024*). Therefore, there is a requirement for a method to improve the resolution of these images for the classification model to work better (*Morrow et al., 2022*; *Ovalle, Vilas & Antelo, 2022*). For this purpose, a form of deep learning technique called Enhanced Super Resolution Generative Adversarial Network (ESRGAN) is used, which specializes in improving the image resolution of an image from lower quality (*Wang et al., 2022b*; *Kandimalla et al., 2022*). Therefore, using ESRGAN on these images could improve the image quality, which can be further used with a deep-learning classification model for better detection accuracy.

VGG-16 is a neural network architecture that has become the winner of the ImageNet challenge (*Mittal, Srivastava & Jayanth, 2022*). It has been trained on 14 million images belonging to 1,000 categories. The trained weights can be used for faster neural network convergence for better detection by training with smaller datasets as well (*Ren & Li, 2022*; *Alaba et al., 2022*). Further, the output layer of the model could be modified to form a bounding box regression (*Wen et al., 2022*). Utilizing this property with the You Only Look Once (YOLO) algorithm can be used for the detection of fishes and identifying species at a large scale (*Bhavya Sree, Yashwanth Bharadwaj & Neelima, 2021*). YOLO works by splitting the image into multiple cells, running the classification model for each such cell, and counting the probability of occurrence; therefore, there exists scope for cross-validating the bounding-box regression with the detected category for better detection accuracy.

The primary contribution of the article includes:

- To create a neural network model composed of VGG-16 with two heads, one for classification and one for regression.
- To combine the regression head of the neural network with a customized YOLO algorithm to find the exact location of the fish.
- To add ESRGAN on low-resolution images to amplify the resolution by four times.

The article is arranged by providing a literature survey in 'Related Works'. The experimental settings are provided in 'Experiment Settings'. The methodology for the experiment is provided in 'Methodology'. The results and observations obtained from the experiment are provided in 'Results'. Other necessary discussions are

provided in 'Discussion'. Finally, 'Conclusion' concludes the paper and suggests future improvement scopes. Portions of this text were previously published as part of a preprint (https://doi.org/10.21203/rs.3.rs-2266266/v2).

## RELATED WORKS

With recent advancements in computational power, artificial intelligence algorithms, and image processing techniques, several newer methods have been proposed for building advanced fish monitoring systems (*Mana & Sasipraba, 2022*; *Hasegawa, Kondo & Senou, 2024*). Input pictures, pre-processing, image segmentation, feature extraction, and image classification are all covered by image processing techniques (*Pauzi et al., 2021*). In general, researchers in various sectors, such as medicine, agriculture, industry, and law enforcement, employ image processing extensively. Digital Image Processing (DIP) is widely used in recognition, remote sensing, image enhancement, color and video processing, and the medical area, among other applications (*Adhikary et al., 2021*). Image processing is also used for visualization, image sharpening, restoration, and image identification (*Bhatt, Naik & Subramanian, 2021*). Blob processing, support vector machine (SVM), neural network, and K-nearest neighbour (KNN) might all be used to classify images (*Machado, Silva & Goldschmidt, 2021*).

High-quality images of fish in complicated habitats are required for effective fish categorisation (*Toğaçar & Ergen, 2022*; *Dai et al., 2024*). The quality of the obtained pictures determines the effectiveness of a fish classification system; however, water turbidity has been a key issue impacting the quality of the acquired images (*Moghimi & Mohanna, 2021*). When the water's turbidity affects the fish's vision, several researchers have proven experimentally that frontal lighting with backlight images can produce relatively acceptable results (*Zheng et al., 2024*). An essential aspect of the machine vision model is pre-processing, which is one of the most critical steps in classifying fish using machine vision models (*Li et al., 2015*; *Yassir et al., 2023*). Image pre-processing includes a variety of processes such as image grayscale, image denoising, image enhancement, image segmentation, and image augmentation (*Dharejo et al., 2024*). The image quality of images acquired from real-world environments can be improved by pre-processing them before the feature extraction process (*Prasetyo, Suciati & Fatichah, 2020*). Image segmentation may be used to discover the area of interest in an image by dividing it into numerous distinct sub-regions (*Yu et al., 2020*).

Various fish farming techniques and methods are currently available (*Prasetyo et al., 2022*). These include everything from large-scale factory farming inside a controlled environment to traditional in-floor floating cage and fence farming to aquaponics and large seine culture, which have both gained popularity in recent years (*Agossou & Toshiro, 2021*; *Ibrahim et al., 2018*). The process of categorizing fish is necessary to precisely measure the behavior of distinct species. Different fish species exhibit more minor variances in size, texture, form, and other physical characteristics than other species of fish (*Jia et al., 2021*; *Horne, Hirst & Atkinson, 2020*) as the three most crucial attributes for visual identification, texture characteristics, shape characteristics, and color characteristics

must all be utilized in conjunction to define image characteristics to produce superior classification results (*Fernandes Junior & Yen, 2019*; *Almero et al., 2020*). Deep learning is a fast-evolving subset of machine learning comprising multiple neural network layers stacked together under different conditions (*Jalal et al., 2020*; *Hasegawa, Kondo & Senou, 2024*). Neural networks have been developed to mimic the human brain's nerve activity for analyzing and learning data like text, sound, images, and more. Deep learning has advanced at a breakneck pace in recent years, and its use in the aquatic industry has become more commonplace (*Deep & Dash, 2019*; *Lu et al., 2024*). Deep learning has also performed well in various applications, including live fish identification, fish classification, behavioral analysis, feeding analysis, biomass estimate, size and weight classification, and more (*Banan, Nasiri & Taheri-Garavand, 2020*). In contrast to typical machine learning methods, the combination of appearance-based characteristics with traditional machine learning techniques is highly interpretable and relatively resilient, and it can obtain excellent results on smaller datasets (*Tamou et al., 2018*). The Visual Geometry Group Network (VGGNet) is a concept established by Oxford University in 2014 to improve visual geometry education (*Chhabra, Srivastava & Nijhawan, 2020*). It has quickly become one of the most popular CNN models, owing to its simplicity and utility (*Agarwal et al., 2021*). By raising the network's depth, VGGNet can increase the model's performance in the context of picture classification (*Nijhawan, 2019*; *Islam et al., 2020*). Instead of using a single convolution layer and a large convolution kernel, multiple convolution layers and smaller convolution kernels are employed (*Thorat, Tongaonkar & Jagtap, 2020*). This significantly reduces the number of parameters while also greatly improving the fitting ability of the network (*Mascarenhas & Agarwal, 2021*; *Schwindt et al., 2024*). VGG-16 is one of the VGGNet variations that have been frequently employed in the categorization of fish species (*Prasetyo, Suciati & Fatichah, 2021*; *Zhao et al., 2019*). Classification for multiple fish species has earlier been performed for fish species of great economic importance, like different variants of carps have been performed with VGG-16 (*Kong et al., 2021*; *Wang et al., 2021b*). The model, on the other hand, is prone to overfitting, and the performance of the classifier is biased in favor of most of the sampled fish species. The researchers have thereby enhanced the effectiveness of CNN by including extra meta-information (such as the migration date and fish length) in CNN's training data. According to many peer-reviewed literature, increasing the number of network layers is helpful in terms of improving classification accuracy (*Voulodimos et al., 2018*; *Chen et al., 2019*). *Rauf et al. (2019)*, *Dhillon & Verma (2020)* developed a fish classification system that is based on a 32-layer VGGNet that is supervised. Even though marine image improvement is vital in marine engineering, more study must be done in this area.

## Motivation for the experiment

After thoroughly investigating the literature, multiple similar works have been observed. Table 1 summarizes the state of the art and describes their limitations addressed in the presented work. Earlier multiple works have been conducted to detect fishes by different methods (*Zhao et al., 2021*; *An et al., 2021*). Some have used motion sensors to sense the waves created by the fishes, some have utilized remote sensing sensors, and some have used

**Table 1  Summary of the State of the Art and their limitations that have been addressed with the proposed work.**

| Source | Objective | Data type | Algorithm | Remark | Limitations |
|---|---|---|---|---|---|
| *Pudaruth et al. (2020)* | Fish species recognition | Camera | Machine learning | 96% accuracy | Lesser number of images & limited scalability |
| *Garcia et al. (2019)* | Fish size recognition | Camera within fishing box | Mask R-CNN | Up to 96% accuracy | Cannot identify species |
| *Hu et al. (2022)* | Fish feeding system by fish movement recognition | Motion sensors | Deep learning | 93.2% accuracy | Cannot work for idle fish |
| *Baker et al. (2022)* | Underwater fish density detection | Camera | Thresholding | 429 fishes were detected | Reduced performance for lower resolution |
| *Palmer et al. (2022)* | Large scale fish monitoring | High resolution camera | Mask R-CNN | 86.10% accuracy | Lower accuracy on low resolution |
| *Palmer et al. (2022)* | Counting fish larvae using smartphones | Smartphone Camera | Faster and Grid R-CNN | 97.3% accuracy | Grown fishes cannot be recognized |
| *Desai et al. (2022)* | Fish species recognition | Camera | ANN | 100% accuracy | Reduced results at lower resolution |
| *Kandimalla et al. (2022)* | Fish detection, classification and counting in fish passages | Camera | Mask R-CNN and YOLO | up to 0.73 mAP | Reduced performance for lower resolution |
| *Lekunberri et al. (2022)* | Identification and measurement of tropical tuna | Camera | Mask R-CNN | 70% accuracy | Only 1 species can be detected which is tuna |
| *Wang et al. (2022a)* | Tracking fish to identify abnormal behaviour in real-time | Camera | YOLOv5 | 76.7% tracking precision | Cannot recognize species |
| *Hong Khai et al. (2022)* | Underwater fish detection and counting | Camera | Mask R-CNN | 97.48% accuracy | Method is species insensitive |
| *Al Smadi et al. (2022)* | Fish classification using deep learning | Camera | Multiple DNN | 98.46% accuracy | Reduced performance at lower resolution |

camera-based approaches (*Franceschelli et al., 2021*; *Li & Du, 2021*). Motion sensors-based method is highly prone to noises in turbulent water (*Li et al., 2021*; *Wageeh et al., 2021*). Remote sensing-based methods perform poorly as most satellites cannot zoom in to levels to detect fish accurately. Therefore, these are primarily useful for detecting zones with a high density of fish (*Belkin, 2021*; *Qiao et al., 2020*). Many works have also been performed to detect fish underwater, but that produces further challenges based on the turbidity of the water (*Ubina & Cheng, 2022*; *Wang et al., 2022*). Other works have been performed to classify fish, including usage of the very high-resolution cameras or pictures captured by cameras dedicated to a limited number of fishes. Therefore, their large-scale implementation is often not feasible (*Zhang et al., 2022*; *Zhang, Chow & Zhang, 2021*). These produce multiple knowledge gaps in existing studies, which can be improved to build a model that can be used to detect and recognize fishes outside water at a considerable scale, verified for a large number of samples, and can be implemented for both high as well as low-resolution images and this motivates to conduct this experiment.

## EXPERIMENT SETTINGS

The work has been conducted on an open-sourced database containing 9,460 images of fish from eight different species and one shrimp species. Within this dataset, 8,109 images were used to train the model, 909 images were used to validate the model during training, and 439 images were used to test the performance. Kodak Easyshare Z650 and Samsung ST60 cameras were used to capture images of the fish. The images were resized to $590 \times 445$ pixels. The species of fish considered for the experiment include gilt-head bream, S*parus aurata*, red sea bream, *Pagrus major*, sea bass, *Centropristis striata*, red mullet, *Mullus barbatus*, horse mackerel, *Trachurus trachurus*, black sea sprat, *Clupeonella cultriventris*, striped red mullet, *Mullus surmuletus*, trout, *Oncorhynchus mykiss* and shrimp, *Caridea*. This is to note that, scientifically, shrimps are not considered fish because they are crustaceans. But they are still included in the study because from an implementational point of view, while catching fish, non-fish species can be caught in the fishing nets as well, and therefore, a method to separate fish and non-fish species is required. These fishes can be replaced with a database of other fish species by transferring the weights and biases of the neural network (*Ulucan, Karakaya & Turkan, 2020*). After the data collection, the images have been further resized for the experiment. Firstly, a dataset was created after resizing all the images to 400 $\times$ 400 pixels (considered as "Original" in this case). Following that, the images were resized to $100 \times 100$ pixels, considered low-resolution images. The experiment was conducted with a Linux operating system having kernel 5.11.0-38-generic, 10th gen intel i5 processor of 4 physical cores with hyperthreading, CUDA enabled NVIDIA GPU with 4GB VRAM, 16GB RAM, and solid state drive based memory.

After this step, the ESRGAN have been used to transform the low-resolution images into super-resolution images, which improved the image pixel density by four times (*Gao, 2021*). This algorithm compares a sequence of low-resolution photos with their corresponding high-resolution images from training data (*Wang et al., 2021a*). The network will learn to translate low-resolution to high-resolution images (LR to HR). The suggested network

is divided into the generative network and the discriminative network (*Rakotonirina & Rasoanaivo, 2020*). We want to train a generation function G, which will convert the LR input photos to the HR images at the end of the process. The ESRGAN model that was used for the experiment was trained on the ImageNet dataset, and further, the trained weights were transferred to our model to make use of the model without retraining.

## METHODOLOGY

The work has been conducted in multiple stages. Firstly, the data collection was performed, followed by pre-processing, application of ESRGAN, VGG-16, and YOLO, and finally, performance comparison. Figure 1 shows the Flowchart of the steps taken to experiment (*Adhikary, 2022*).

### Neural network architecture

The detection of the fishes and their species has been facilitated using a two-headed neural network model on top of a VGG-16 model (*Rabbi et al., 2020*). The two heads of the network perform different tasks. Where one head is used to perform a bounding box regression, which detects the exact location of the fish within the image and on the other hand, and the other head of the network is used to perform the categorical classification, which is used to detect the species of the fish (*Gu et al., 2021*). The most prominent difference between the two architectures is the input and output shapes of all the layers till the categorical classification and bounding box regression have been performed. Between these layers, the input/output shapes of the ESRGAN-based network were roughly four times larger than the low-resolution images. The VGG-16 head is made by taking the weights of the model in non-trainable fashion, then flatenning them. Later, a trainable multi-layer perceptron head was added with a dense layer of 512 nodes with rectified linear unit (ReLU), followed by a dropout of 0.5, then a dense layer of 512 nodes with relu and a dropout of 0.5 and finally dense layer of nine nodes with softmax activation function to produce categorical outputs for the different classes of fish images. This head was trained with categorical crossentropy loss. On the other hand, the bounding box regression head was made with a dense layer of 128 nodes and relu activation with 0.5 dropout rate, followed by a dense layer of 64 nodes with relu and 0.5 dropout, then 32 node layer with relu and 0.5 dropout and finally a dense layer of four nodes with relu activation. These four layers indicates the top-left, top-right, bottom-left and bottom-right corners of the bounding box. This head was trained with mean squared error loss. For both the model, adam optimized with 0.0001 learning rate was used for the training. The model was trained for 50 epochs with each epoch having 50 steps each. A batch size of 32 was used for training. A a k-fold cross validation with 10 splits were used to train and validate the model. Figure 2 shows the picture of the proposed neural architecture with two heads to perform both classification and bounding box regression simultaneously.

### Combining YOLO and VGG-16

An important deep learning-based object detection algorithm is called YOLO (*Zhang et al., 2018*). This works by splitting the target image into multiple smaller segments and applying

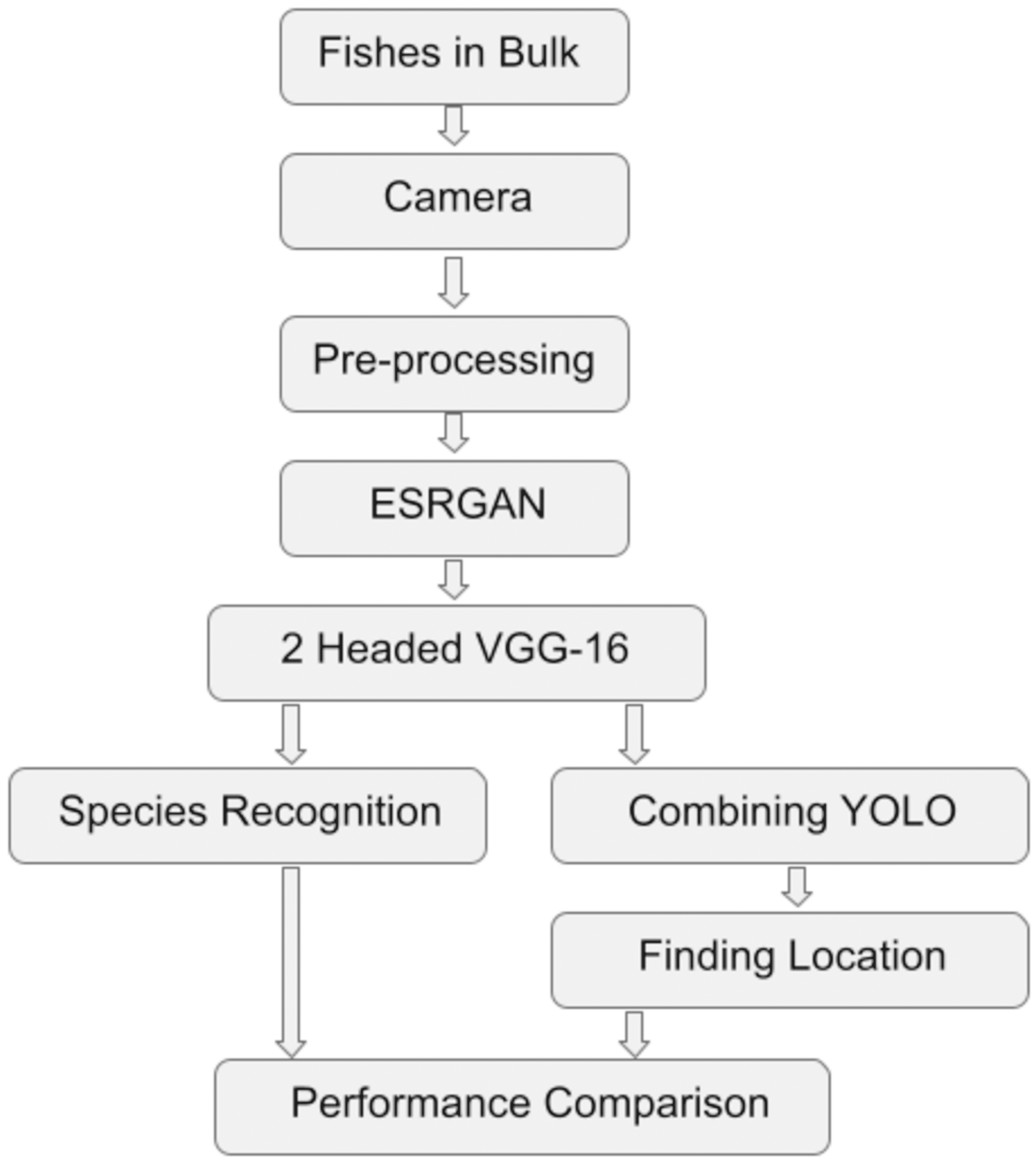

**Figure 1** The flowchart of the procedures undergone to experiment.

a classification algorithm to each of these segments. A corresponding matrix is generated, where each element is associated with a segment of the image, and the magnitude is based on a corresponding class as detected by the classification model. Following this, probability mapping is performed to detect multiple objects within the same image. We have modified the usage of a regular YOLO algorithm to suit our experiment better (*Zhao et al., 2017*). We have used the earlier defined ESRGAN-based VGG-16 model on each smaller segment of the image, and based on the edges of the bounding box regression, the next segment of the image has been considered, which ultimately provides the location of the entire fish along with its species.

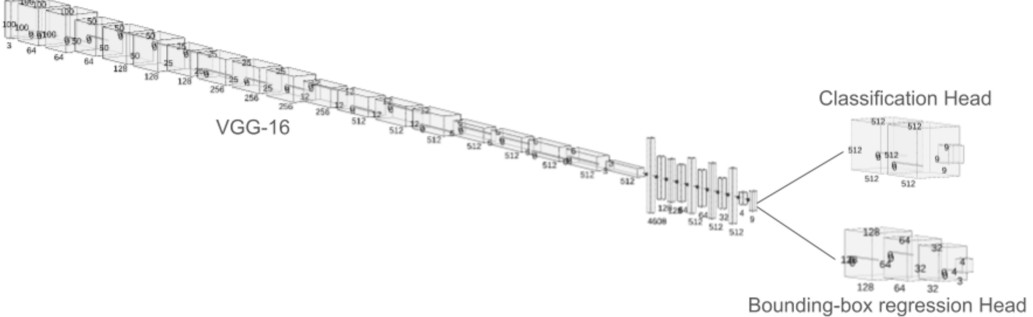

**Figure 2** The proposed two-headed neural network model to perform both classficiation and bounding-box regression at the same time.

## Innovative combination of YOLO and VGG-16 with dual-headed neural network

The primary novelty of the work involves merging a dual-headed neural network along with VGG-16 based YOLO that can perform both classification and bounding box regression on a particular tile concurrently and accurately. This is done by first using the neural network defined earlier that takes in images as input, performs classification and bounding box regression with its two different heads and outputs the results. Now, this is done by first increasing the resolution of the image to four times using ESRGAN, then splitting the full image into multiple blocks and running the neural network on each tile. Finally YOLO is used to adjust the overlaps in bounding box regression to properly localize the bounding boxes. This enables the model to take advantage of the low latency bounding box regression, accurate classification of VGG-16, 4x image resolution enhancement of ESRGAN and reliability of YOLO algorithm.

## Performance comparison

The performance of the work has been compared with several metrics. The detection performance was measured with accuracy, precision, recall, training, and testing time. Following this, the difference between the original image and ESRGAN-generated images were compared based on several metrics, which include mean squared error (MSE), root mean squared error (RMSE), peak signal-to-noise ratio (PSNR), structural similarity index (SSIM), universal image quality index (UQI), multi-scale structural similarity index (MSSSIM), erreur relative globale adimensionnelle de synthese (ERGAS), spatial correlation coefficient (SCC), relative average spectral error (RASE) and spectral angle mapper (SAM) (*Hu et al., 2020*; *Liang & Weller, 2016*).

## Ethics statement

The author confirms that the ethical policies of the journal, as noted on the journal's author guidelines page, have been adhered to. No ethical approval was required as the data used in the experiment was obtained from published literature (*Ulucan, Karakaya & Turkan, 2020*).

## Low Resolution

## Super Resolution

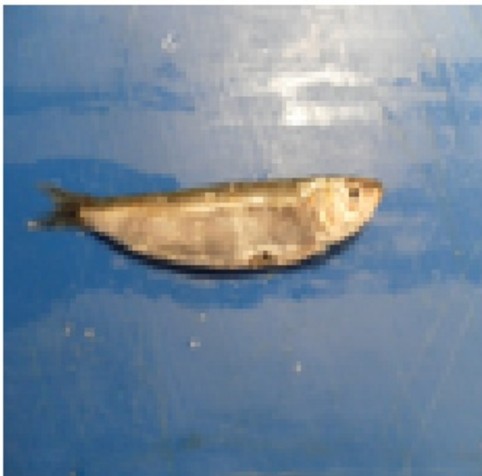
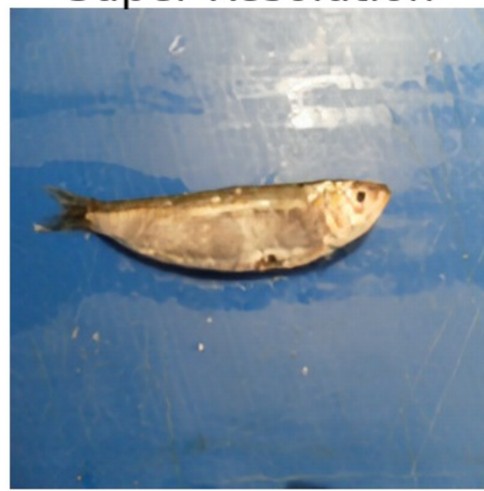

**Figure 3** **Difference between a low-resolution fish image and its corresponding super-resolved image generated by ESRGAN.**

# RESULTS

Different patterns were observed while experimenting. The following texts summarize the most prominent results obtained from the experiment.

## Comparison of original and ESRGAN generated super-resolved images

The application of ESRGAN on low-resolution images provides a four times clearer picture, which can be used for more accurate fish detection and species identification. When caught in bulk, usually on a ship, the fish are kept on the deck or similar associated areas where the fish are further sorted. As a large volume of fish is kept, capturing individual pictures of the fish makes it difficult to identify the species. Hence, while capturing photos of a large number of fish, the resolution of each fish is generally very low. The low-resolution image symbolizes these small segments of images containing individual fish. Figure 3 shows the difference between a symbolic representation of a cropped low-resolution image captured by a camera at a deck of a ship from a distance and an ESRGAN generated super-resolved image of the fish. At first glance, the differences between the two are visible. The low-resolution image appears blurry, but the super-resolution image is much sharper.

Further, more differences are observed by comparing the original 400 × 400 image (not the low-resolution image) and the super-resolution image based on different statistical parameters. Table 2 records the average statistical differences between the original and super-resolved images for all sample images. It has been observed that the MSE of black sea sprat was the lowest among all 20.12 and the highest for horse mackerel, which was 87.9. This indicates the increasing difficulty for the ESRGAN algorithm to super-resolve the images. This has supposedly occurred because of these fish species' growing color contrast and more complex texture. Accordingly, the RMSE score also follows the order of black

**Table 2  The image restoration quality comparison for the original and super-resolution images (Units not mentioned as all comparisons are based on unitless quantities).**

| Species | MSE | RMSE | PSNR | SSIM | UQI | MSSSIM | ERGAS | SCC | RASE | SAM |
|---|---|---|---|---|---|---|---|---|---|---|
| Black Sea Sprat | 20.12 | 4.48 | 35.09 | 0.9204,0.9206 | 0.998 | 0.9694 | 1,207.38 | 0.1065 | 150.92 | 0.0251 |
| Gilt Head Bream | 55.11 | 7.42 | 30.71 | 0.8698,0.8714 | 0.989 | 0.9478 | 1,943.06 | 0.0947 | 242.88 | 0.0511 |
| Horse Mackerel | 87.90 | 9.37 | 28.69 | 0.8278,0.8287 | 0.991 | 0.9242 | 2,865.90 | 0.1003 | 358.23 | 0.0641 |
| Red Mullet | 76.96 | 8.77 | 29.26 | 0.8418,0.8422 | 0.995 | 0.9303 | 2,399.61 | 0.1061 | 299.95 | 0.053 |
| Red Sea Bream | 47.96 | 6.92 | 31.32 | 0.8704,0.8709 | 0.995 | 0.9481 | 1,887.01 | 0.1018 | 235.87 | 0.0433 |
| Sea Bass | 86.94 | 9.32 | 28.73 | 0.8349,0.8354 | 0.993 | 0.9304 | 2,497.43 | 0.1024 | 312.17 | 0.0656 |
| Shrimp | 49.51 | 7.03 | 31.18 | 0.8827,0.8831 | 0.996 | 0.9534 | 1,813.61 | 0.1095 | 226.70 | 0.0396 |
| Striped Red Mullet | 63.43 | 7.96 | 30.10 | 0.8514,0.8517 | 0.995 | 0.9377 | 2,136.50 | 0.1019 | 267.06 | 0.0487 |
| Trout | 80.25 | 8.95 | 29.08 | 0.8542,0.8566 | 0.987 | 0.9396 | 2,530.33 | 0.1423 | 316.29 | 0.0631 |

sea sprat, red sea bream, shrimp, gilt-head bream, striped red mullet, red mullet, trout, sea bass, and horse mackerel. PNSR, or peak signal-to-noise ratio, indicates the degree of improvement of the resolved image compared to the original image. Higher PNSR scores indicate a better restoration. Accordingly, in this case, the same order has been found for all the fish species. The UQI or universal image quality index represents the summation of errors between the restored and original image based on loss of correlation, luminance distortion, and contrast distortion. The UQI values have been found to be maximum in Black Sea sprat and minimum in trout. For the multi-scale structural similarity index (MSSSIM). However, a direct trend cannot be observed; an indirect downward trend of fluctuations could be kept in the order mentioned earlier. An upward trend can be observed for ERGAS values, indicating an increasing computational complexity. The RASE or relative average spectral error for the fishes was found to be in a similar order with slight fluctuations. The SAM or spectral angle mapper error for the different fishes shows an upward trend with minor variations.

Investigating further, the color histogram for all fish species has been evaluated for original, low-resolution, and ESRGAN-generated high-resolution images, shown in Fig. S1. This figure contains a pixel density distribution histogram for red, green, blue, and all colors combined (represented by yellow color) for pixel intensity ranging from 0–255. For all the species, it can be observed that the peaks of all the graph colors for low-resolution images are uneven. Still, for the original and ESRGAN-generated images, the peaks are smoother. Also, the ESRGAN-generated images and original images have a higher number of pixels for each intensity level for each color compared to the low-resolution image, which is evident from the fact that the low-resolution image had been reduced by $1/4^{th}$ of the original image. However, original and ESRGAN-generated super-resolved images have no generic differences, but some variations can still be observed, which change according to the target picture. For example, in graphs of Horse Mackerel, during the intensity range of 225 to 250, few blue bars are visible in the original image, which is missing in ESRGAN-generated images. Similar patterns can also be observed in many other fish species as well.

| | | Predicted | | | | | | | | |
|---|---|---|---|---|---|---|---|---|---|---|
| | | Black Sea Sprat | Gilt Head Bream | Horse Mackerel | Red Mullet | Red Sea Bream | Sea Bass | Shrimp | Stripped Red Mullet | Trout |
| **Actual** | **Black Sea Sprat** | **0.97** | 0.00 | 0.01 | 0.01 | 0.01 | 0.00 | 0.01 | 0.00 | 0.00 |
| | **Gilt Head Bream** | 0.00 | **0.96** | 0.01 | 0.00 | 0.00 | 0.00 | 0.01 | 0.01 | 0.00 |
| | **Horse Mackerel** | 0.00 | 0.00 | **0.98** | 0.00 | 0.01 | 0.00 | 0.00 | 0.00 | 0.00 |
| | **Red Mullet** | 0.00 | 0.00 | 0.00 | **0.98** | 0.00 | 0.00 | 0.00 | 0.00 | 0.00 |
| | **Red Sea Bream** | 0.00 | 0.01 | 0.00 | 0.01 | **0.96** | 0.01 | 0.00 | 0.00 | 0.02 |
| | **Sea Bass** | 0.01 | 0.01 | 0.00 | 0.00 | 0.00 | **0.97** | 0.00 | 0.00 | 0.01 |
| | **Shrimp** | 0.01 | 0.01 | 0.01 | 0.00 | 0.00 | 0.00 | **0.98** | 0.00 | 0.00 |
| | **Stripped Red Mullet** | 0.00 | 0.01 | 0.00 | 0.00 | 0.01 | 0.01 | 0.00 | **0.98** | 0.00 |
| | **Trout** | 0.00 | 0.00 | 0.00 | 0.00 | 0.01 | 0.01 | 0.00 | 0.00 | **0.96** |

**Figure 4** Confusion matrix for the detection shows reliable and consistent results across all species.

## Detection performance

After applying ESRGAN to the images and obtaining the generated super-resolution images, the photos were passed into the earlier discussed neural network and the YOLO algorithm. This results in the detection of each fish along with the species. Figure 4 shows the confusion matrix of the detection and Fig. 5 shows a sample detection on the super-resolved images where each of the species of fish (written within a black patch with white fonts) has been detected along with a precise surrounding bounding box marked with a green line. After running the algorithm on all test images, a detection accuracy 96.5% has been obtained. The details have been recorded in Table 3. Further, it has been observed that the precision of the performance was 0.93, and the recall of the performance was 0.96, which indicates that there were fewer false-negative errors compared to false-positive errors. The model took 1,323.8 s to train and 12.6 s to generate all the results. Further, from the confusion matrix, it can be seen that for all the species, the detection is successful at a range of 96–98% accuracies. Thus, it can be claimed that for all the fish species, the detection results are consistent and this further clarifies the reliability of the model.

A comparison of the proposed model with other super-resolution models has been presented in Table 4. From this table, it is clear that the proposed model surpasses the performances obtained from BiCubic, Waifu2x, Upscayl, RealScaler, *Toğaçar & Ergen (2022)*, *Moghimi & Mohanna (2021)*, *Zheng et al. (2024)* and *Dharejo et al. (2024)* in terms of accuracy by each respectively attaining accuracy of 92.5%, 89.9%, 92.7%, 88.4%, 91.8%, 90.5%, 92.1% and 88.2%, respectively. Similarly, the comparison of the proposed model with other transfer learning models has also shown similar results. VGG-19, XCeptionV3

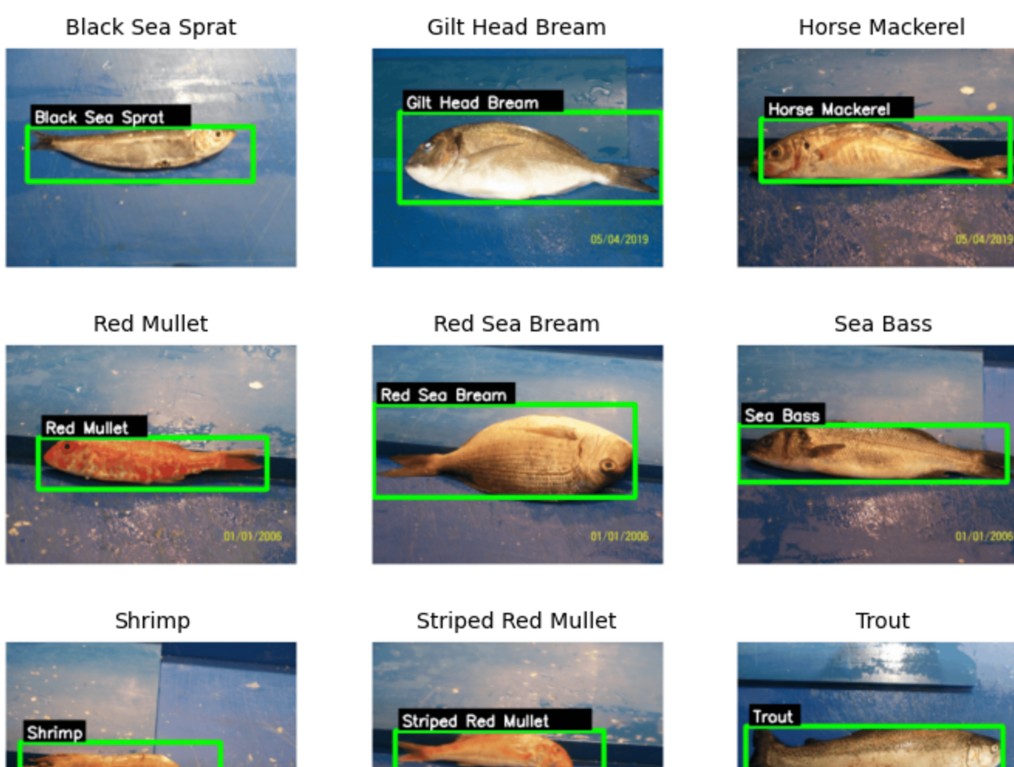

**Figure 5** Detection results for the algorithm including the species name and its precise location marked by bounding box.

**Table 3** Performances for the detection with ESRGAN-based approach.

| Algorithm | Accuracy (%) | Precision | Recall | F1 score | AUC score | Train time (s) | Test time (s) |
|---|---|---|---|---|---|---|---|
| Proposed | 96.5 | 93.2 | 96.1 | 94.6 | 98.5 | 1,323.8 | 12.6 |
| YOLO | 87.4 | 85.3 | 86.1 | 85.6 | 89.1 | 1,285.4 | 13.2 |
| SSD | 91.2 | 90.7 | 92.1 | 91.3 | 93.2 | 2,342.6 | 26.3 |
| ResNet | 88.4 | 89.2 | 87.4 | 88.2 | 91.4 | 1,865.3 | 21.4 |
| *Knausgård et al. (2022)* | 92.4 | 90.1 | 91.4 | 90.7 | 94.2 | 1,643.1 | 14.2 |
| *Ovalle, Vilas & Antelo (2022)* | 89.8 | 87.4 | 88.1 | 87.7 | 91.7 | 1,594.3 | 16.4 |
| *Kandimalla et al. (2022)* | 90.7 | 88.1 | 87.6 | 87.8 | 92.9 | 1,586.4 | 15.2 |
| *Alaba et al., (2022)* | 87.6 | 85.2 | 84.2 | 84.6 | 89.1 | 1,891.1 | 22.5 |
| *Mana & Sasipraba (2022)* | 90.1 | 88.4 | 87.9 | 88.1 | 93.5 | 2,153.4 | 21.3 |

**Table 4   Performances for the detection with different super resolution techniques.**

| Algorithm | Accuracy (%) | Precision | Recall | F1 score | AUC score | Train time (s) | Test time (s) |
|---|---|---|---|---|---|---|---|
| Proposed | 96.5 | 93.2 | 96.1 | 94.6 | 98.5 | 1,323.8 | 12.6 |
| BiCubic | 92.5 | 87.3 | 89.5 | 88.3 | 94.3 | 1,352.7 | 14.1 |
| Waifu2x | 89.9 | 88.1 | 91.7 | 89.8 | 93.6 | 1,521.8 | 17.4 |
| Upscayl | 92.7 | 90.1 | 90.6 | 90.3 | 94.2 | 1,893.2 | 18.3 |
| RealScaler | 88.4 | 89.2 | 85.3 | 87.2 | 91.3 | 1,534.8 | 22.6 |
| *Toğaçar & Ergen (2022)* | 91.8 | 90.2 | 90.4 | 90.3 | 94.0 | 1,786.3 | 20.4 |
| *Moghimi & Mohanna (2021)* | 90.5 | 87.5 | 86.2 | 86.8 | 91.8 | 1,923.3 | 18.5 |
| *Zheng et al. (2024)* | 92.1 | 89.2 | 90.1 | 89.6 | 95.1 | 1,854.3 | 19.2 |
| *Dharejo et al. (2024)* | 88.2 | 87.1 | 86.6 | 86.8 | 90.5 | 2,132.5 | 21.5 |

**Table 5   Performances for the detection with different transfer learning techniques.**

| Algorithm | Accuracy (%) | Precision | Recall | F1 score | AUC score | Train time (s) | Test time (s) |
|---|---|---|---|---|---|---|---|
| Proposed | 96.5 | 93.2 | 96.1 | 94.6 | 98.5 | 1,323.8 | 12.6 |
| VGG-19 | 95.2 | 94.3 | 95.2 | 94.7 | 97.4 | 2,142.4 | 15.2 |
| XCeptionV3 | 92.1 | 92.7 | 91.0 | 91.8 | 94.6 | 1,543.7 | 18.3 |
| InceptionV3 | 93.7 | 91.8 | 92.4 | 92.0 | 95.3 | 1,634.8 | 16.6 |
| *Hasegawa, Kondo & Senou (2024)* | 94.2 | 92.3 | 93.5 | 92.9 | 97.1 | 1,643.7 | 17.2 |
| *Lu et al. (2024)* | 91.6 | 90.5 | 91.2 | 90.8 | 94.2 | 1,862.6 | 18.2 |
| *Schwindt et al. (2024)* | 90.3 | 91.2 | 88.6 | 89.8 | 92.5 | 1,768.1 | 24.2 |
| *Dai et al. (2024)* | 91.2 | 90.2 | 88.7 | 89.4 | 91.9 | 1,938.6 | 22.4 |

and InceptionV3 have obtained 95.2%, 92.1% and 93.7%, respectively. For all of these cases, the proposed model have attained up to 96.5 accuracy. Thus, this further confirms the reliability of the model. Details of which have been added in Table 5.

The performances of the proposed model have been thoroughly investigated with 10-fold cross-validation method. During all the 10 iterations, a range of accuracies was observed, which was between 94.1%–98.9%. The mean accuracy was found to be 96.5%. The standard deviation of accuracy during the 10-fold cross-validation was found to be 1.68, thus further strengthening the findings. The details of the cross-validation, which includes accuracy, precision, recall, f1-score, AUC score, training time and testing time, can be found in Table 6. A consistent result can be observed during all iterations.

## Comparison with the state-of-the-art methods

The presented work has outperformed the state-of-the-art methods in multiple parameters. The technique shown by *Pudaruth et al. (2020)* has been surpassed by testing with a larger dataset. Also, the presented experiment is more scalable because of deep learning. The work presented by *Garcia et al. (2019)* cannot differentiate between different species of fish, and therefore, the presented model surpasses this study by introducing multiple species detection models. The experiment presented by *Hu et al. (2022)* cannot be used to detect the exact location and species of fish that the proposed model has performed. The

**Table 6  10-fold cross validation performance for the proposed model shows consistent results across all iterations.**

| Fold iteration | Accuracy (%) | Precision | Recall | F1 Score | AUC score | Train time | Test time |
|---|---|---|---|---|---|---|---|
| 1 | 98.5 | 95.3 | 96.8 | 96.0 | 99.6 | 1,376.8 | 12.9 |
| 2 | 96.3 | 91.9 | 97.2 | 94.4 | 98.5 | 1,434.2 | 12.2 |
| 3 | 97.6 | 92.6 | 96.5 | 94.5 | 99.8 | 1,443.4 | 11.7 |
| 4 | 98.9 | 95.3 | 95.1 | 95.1 | 99.1 | 1,216.7 | 12.6 |
| 5 | 95.6 | 92.3 | 96.6 | 94.4 | 98.2 | 1,252.7 | 13.5 |
| 6 | 97.2 | 95.1 | 98.2 | 96.6 | 98.5 | 1,269.3 | 12.8 |
| 7 | 97.4 | 90.1 | 95.2 | 92.5 | 98.6 | 1,321.9 | 11.7 |
| 8 | 95.3 | 93.7 | 96.1 | 94.8 | 98.7 | 1,268.3 | 11.5 |
| 9 | 94.2 | 93.6 | 94.2 | 93.8 | 97.3 | 1,301.5 | 14.6 |
| 10 | 94.1 | 92.1 | 95.1 | 93.5 | 96.7 | 1,353.2 | 12.6 |
| Mean | 96.5 | 93.2 | 96.1 | 94.6 | 98.5 | 1,323.8 | 12.6 |
| STD | 1.68 | 1.71 | 1.19 | 1.16 | 0.94 | 76.9 | 0.93 |

model built by *Baker et al. (2022)* could not detect the species, has reduced performance at a lower resolution, and cannot pinpoint the exact location of the fish. The presented model has addressed all of these parameters. *Palmer et al. (2022)*; *Al Smadi et al. (2022)* had performance degradation at lower resolution images where the method proposed in the experiment obtained higher accuracy, which is 96.5% for lower resolution. *Palmer et al. (2022)* had built a model that cannot individually identify the fish species. However, although the method proposed in our experiment has not been tuned in for detecting fishes at the larval stage, grown-up fishes could be identified. The model developed by *Desai et al. (2022)* has yet to be tuned to recognize fish species at a lower resolution, which our method has solved. The algorithm of *Kandimalla et al. (2022)* needs to be tuned appropriately to handle lower-resolution images, which our algorithm has solved. Our method has outperformed (*Lekunberri et al., 2022*; *Wang et al., 2022a*) by both performance accuracy and the number of species detected. The model shown in *Hong Khai et al. (2022)* could not identify different species that our proposed model has performed.

## DISCUSSION

The results that have been obtained are further compared with other strategies and the state of the art. Following this, the model has been retested with low-resolution images to compare the performance difference, and the performances are recorded in Table 7. Also, the importance of the work, its practical significance, and implementations. have been further discussed in the following text.

### Quality control documentation

The study was conducted on eight fish species and one shrimp species, with approximately one thousand images from each species. The detection performances obtained during the experiment can vary with changing the images, but a properly tuned and trained model with similar data may consistently provide accuracy over 85%. The resource consumption

**Table 7 Performances for the detection with raw low-resolution data to better compare the performance.**

| Method | Accuracy (%) | Precision | Recall | F1 score | AUC score | Train time (s) | Test time (s) |
|---|---|---|---|---|---|---|---|
| Proposed | 96.5 | 93.2 | 96.1 | 94.6 | 98.5 | 1,323.8 | 12.6 |
| Raw | 82.3 | 81.5 | 83.2 | 82.3 | 85.8 | 738.2 | 9.2 |

for building the model can also change depending on the underlying hardware. The presence of other processes or threads running in the background can further degrade the model's resource consumption; therefore, hardware with minimal background processes and threads is required to replicate results similar to the experiment. Minor variations in the accuracy may be observed when the model is applied to real-world data, as there might be different backgrounds compared to the ones used during training. Also, other lighting conditions, foggy weather conditions, and contamination of foreign materials may further introduce accuracy variations. A very close overlap of two fishes of the same species may also sometimes be detected as a single fish. Despite these challenges, the model can still work better than most other algorithms because the unpredictability of any threshold-based algorithm is eliminated by using smart deep learning algorithms, and the image quality for the purpose has been improved by another innovative deep learning algorithm, which in this case is ESRGAN.

## Real world implementation & practical feasibility

The method can be best utilized in the real world at fish sorting centers where a large volume of fish needs to be sorted based on their species and sizes. The sorting based on size can easily be performed by reading the weight of the fish, but the species identification at a fast scale for a large volume would require a vision-based system where the presented method can be utilized. To build such a system, a properly calibrated pick-and-place machine on a conveyor belt must be integrated with a camera and processing unit. The conveyor belt would carry many fish where the camera would grab the image, the processing unit would find the species and location of that specific fish, and the pick-and-place machine would pick the target fish and place it in its appropriate box. Unlike other methods where one fish must be processed at a time, the presented method can sort a large number of fish simultaneously.

## Priorities for next steps & possible impacts in next 10 years timescale

Certain steps need to be taken to complete this work and make it deployable. Firstly, during the first year, the model needs to be tested for more species of fish, and as discussed earlier, a pick-and-place machine needs to be integrated with the system for better sorting. Then, appropriate calibration of the model is necessary to make the system robust in an uncontrolled environment. Once the calibration is properly executed, during the next 5 years, the entire system could be processed to be deployed in different places where fish sorting is necessary, such as in fishing ships or fish markets. Once the system has been deployed, many fish could be automatically sorted based on their species. Accordingly,

manual labor involvement in sorting these fish would be dramatically reduced, and the sorting speed would exponentially increase. This would also reduce the cost of handling all these fish. Sorting smaller fish generally requires a lot of focus for manual laborers, which the proposed system could completely or partially replace.

### Limitations of the work

The work is exploratory and has been conducted on eight fish and one shrimp species. This number needs to be higher for a real-world application. The work needs to be extended by including several other fish species. The work has been tested on images of 500 fish from each category, and this is also a smaller number to have a concrete result. The training has been performed for images under similar lighting conditions and environments; therefore, more variations of environment and light conditions are required for training the model. The classification and regression model is based on the VGG-16 network trained on ImageNet weights. Although VGG-16 provides very accurate results, the network itself is extensive, and therefore, fitting the model into storage constraint devices produces deployment challenges that can be further solved by introducing better alternatives.

## CONCLUSION & FUTURE SCOPES

The fish detection and recognition models are fundamental in large-scale fish industries for quickly sorting fish according to their appropriate species. Existing models have many scalability, feasibility, and performance limitations, which have been solved with the presented work. The work uses VGG-16 by customizing and adding two heads with it, one for classification and one for bounding box regression, and the bounding box regression has been integrated with a customized YOLO algorithm for detecting the precise location of the fish in low latency. Also the algorithm uses ESRGAN to amplify the resolution of the images four times, further enhancing the detection performance.

The proposed experiment was conducted on 9,460 images for eight different species of fish and one species of shrimp and obtained up to 96.5% detection accuracy. The model has further been tested for comparison with raw data, which had obtained 82.3% detection accuracy, indicating the improvement of the proposed algorithm. Investigating the super-resolution images, it has been observed that the black sea sprat had the lowest MSE of 20.12, and red mullet trout had the highest MSE of 80.25 among all species when the super-resolution images were compared to the original images. Also, the color pixel density distribution histogram revealed relations between original, low-resolution, and ESRGAN-generated super-resolution images.

The work can further be improved by increasing the number of species and testing the model on a larger scale. Integrating the system with a pick-and-place machine with proper calibration would complete the technology, and a very fast, accurate, very large-scale fish sorting machine could be built.

## REPRODUCIBILITY

- The dataset have been obtained from this article: DOI: 10.1109/ASYU50717.2020. 9259867.

- Computing infrastructure: The experiment was conducted with a Linux operating system having kernel 5.11.0-38-generic, 10th gen intel i5 processor of four physical cores with hyperthreading, CUDA enabled NVIDIA GPU with 4GB VRAM, 16GB RAM, and Solid State Drive based memory.

## DESCRIPTION OF MODELS USED

The experiment was conducted by first using the Enhanced Super Resolution Generative Adversarial Neural Network (ESRGAN) model. Later a custom neural network was developed with a VGG-16 base and two heads, one capable of classification and another capable of creating bounding box regression.

### Funding
The authors received no funding for this work.

### Competing Interests
The authors declare that they have no competing interests.

Subhrangshu Adhikary is a Director for Spiraldevs Automation Industries Pvt. Ltd. Saikat Banerjee is a Director for Aerosys Defence and Aerospace Pvt. Ltd.

### Author Contributions
- Subhrangshu Adhikary conceived and designed the experiments, performed the experiments, analyzed the data, performed the computation work, prepared figures and/or tables, authored or reviewed drafts of the article, and approved the final draft.
- Saikat Banerjee conceived and designed the experiments, performed the experiments, analyzed the data, performed the computation work, prepared figures and/or tables, authored or reviewed drafts of the article, and approved the final draft.
- Rajani Singh conceived and designed the experiments, performed the experiments, analyzed the data, performed the computation work, prepared figures and/or tables, authored or reviewed drafts of the article, and approved the final draft.
- Ashutosh Dhar Dwivedi conceived and designed the experiments, performed the experiments, analyzed the data, performed the computation work, prepared figures and/or tables, authored or reviewed drafts of the article, and approved the final draft.

### Data Availability
The code is available in the Supplemental Files.

### Supplemental Information
Supplemental information for this article can be found online at http://dx.doi.org/10.7717/peerj-cs.2860#supplemental-information.

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
