# Peer review of "Fish species identification on low resolution—a study with enhanced super-resolution generative adversarial network (ESRGAN), YOLO and VGG-16"

_PeerJ Computer Science, doi:10.7717/peerj-cs.2860_

## Round 0.1 · original submission · Major Revisions

The review process is now complete. While finding your paper interesting and worthy of publication, the referees and I feel that more work could be done before the paper is published. My decision is therefore to provisionally accept your paper subject to major revisions. More details are needed.

Reviewer 1 ·

Basic reporting

In this paper, the authors propose a transfer learning-based method for fish species identification under low resolution. However, there are major concerns as follows:

- It is important for the readability of the paper that the references are listed in order. This will also improve readability. Additionally, it would be better if citations are given in the format [5] instead of (5).

-The literature review presented by the authors appears to be incomplete. It is important to include studies from the years 2023-2024 to better understand the current state of the topic.

Experimental design

- The authors list their contributions to the literature in the form of bullet points. However, there are some issues at this stage. For instance, items like 'To complete the detection model and compare the performances' and 'To detect fish and recognize the species at a large volume with low-resolution images' cannot be considered as contributions. The contributions should highlight how the study differentiates from similar works. Upon reviewing the other points too, this aspect is found to be lacking. What is the contribution of the current study? Is it the first study to use the YOLO model or the first to enhance resolution?

- In order to understand the model structure of the CNN networks used, it is important to present it in a detailed and explanatory manner, as seen in similar papers, rather than the format presented in Figure 2. This is a crucial step for the reproducibility of the model.

-One of the most important steps in training CNN networks is determining the correct hyperparameters. It is crucial for reproducibility that the authors provide details on how they trained the CNN networks used in the paper. For example, batch size, learning rate, number of epochs, and the optimizer used, etc.

Validity of the findings

- The major shortcoming of the study lies in the evaluation of its contribution to the literature. For a detailed analysis of the proposed model's performance, it is essential to compare the model's performance (accuracy, recall, F1, etc.) with other similar studies.

- Using the AUC metric would also be beneficial for better understanding the contribution of the study.

-The choice of training-test split during model training directly impacts the model's performance, as the achieved model success may be solely related to this test splitting. Therefore, using k-fold cross-validation would be beneficial for better evaluating the model's performance. Providing the results of different folds in terms of average accuracy and standard deviation would help in better assessing the model's performance.

Reviewer 2 ·

Basic reporting

The overall structure of the manuscript is disorganized. Section 3.1 should be the experiment setting under the experiments section. Additionally, Section 5.1, which compares the proposed method with state-of-the-art methods, should not be placed in the Discussion section (Section 5).

Experimental design

1)The manuscript introduces two major innovative contributions: ESRGAN and the modified YOLO algorithm. However, there is a lack of detailed description in the METHODOLOGY (Section 3) regarding these innovations.
2)The experimental section lacks sufficient data. There is no comparative data from other methods, which undermines the claim that the proposed method outperforms others. The manuscript should added results from other methods for a more convincing argument.

Validity of the findings

1)Table 2 needs to be reformatted to ensure that all text in each column is fully visible.
2)The manuscript presents a conflict between the MSE and PSNR metrics for the super-resolution of black sea sprat images. Specifically, while the higher MSE indicates higher difficulty for the ESRGAN to super-resolve the images, the higher PNSR indicates a better restoration. This discrepancy requires a reasonable explanation.

Reviewer 3 ·

Basic reporting

You can add a paragraph containing the organization of the article to the introduction section.
Is Figure 1 correct? The 2-head VGG network recognizes species from images and also locates fish using it with YOLO?
Shouldn't the species be identified after first locating the fish using YOLO?
References must be in order.

Experimental design

The material and method section is weak. It should be developed.
VGGNet architecture and YOLO should be explained more descriptively.
Describe the hyper-parameters used in your proposed model.
What did you use for hyper-parameter optimization? Which activation function, which optimizer, or which learning rate? How did you decide on these?
Compare your results with similar studies using table.

Validity of the findings

Evaluation metrics should be explained with formulas.
You should compare the accuracy value you obtained with the ESRGAN method with different methods and different hyperparameters.
You only gave the result of the method you suggested. Perhaps different transfer learning architectures will be more successful. We don't know this.
Add values such as F-score and AUC.
Also show your results with graphs. For example, ROC curve, Confusion matrix

---

## Round 0.2 · Major Revisions

I have completed my evaluation of your manuscript. The reviewers recommend reconsideration of your manuscript following major revision. I invite you to resubmit your manuscript after addressing the comments below.

Reviewer 1 ·

Basic reporting

No comment

Experimental design

- The authors were asked to compare the numerical results of the current model with similar studies in the literature to evaluate its contribution. However, in Tables 3, 4, 5, and 6, they have only compared the current model with other traditional models. This is an incomplete revision, and the study's contribution to the literature remains unclear.

- The authors were also asked to perform a k-fold procedure and present results such as training accuracy with the mean and standard deviation obtained from each fold. However, they have only verbally stated that k-fold was used, without providing the requested results, which does not adequately address the revision request.

Validity of the findings

No comment

Additional comments

No comment

---

## Round 0.3 · accepted · Accept

Since the comments have been addressed, we are happy to inform you that your manuscript has been accepted for publication.

Reviewer 1 ·

Basic reporting

The authors have completed the revision process successfully. The manuscript can be accepted.

Experimental design

The authors have completed the revision process successfully. The manuscript can be accepted.

Validity of the findings

The authors have completed the revision process successfully. The manuscript can be accepted.